# Drone-Based Identification and Monitoring of Two Invasive Alien Plant Species in Open Sand Grasslands by Six RGB Vegetation Indices

László Bakacsy [1],* , Zalán Tobak [2] , Boudewijn van Leeuwen [2] , Péter Szilassi [2] , Csaba Biró [3] and József Szatmári [2]

1 Department of Plant Biology, University of Szeged, Közép Fasor 52, H-6727 Szeged, Hungary
2 Department of Geoinformatics, Physical and Environmental Geography, University of Szeged, Egyetem utca 2-6, H-6722 Szeged, Hungary
3 Kiskunsag National Park Directores, Liszt Ferenc utca 19, H-6000 Kecskemét, Hungary
* Correspondence: bakacsy@bio.u-szeged.hu

**Abstract:** Today, invasive alien species cause serious trouble for biodiversity and ecosystem services, which are essential for human survival. In order to effectively manage invasive species, it is important to know their current distribution and the dynamics of their spread. Unmanned aerial vehicle (UAV) monitoring is one of the best tools for gathering this information from large areas. Vegetation indices for multispectral camera images are often used for this, but RGB colour-based vegetation indices can provide a simpler and less expensive solution. The goal was to examine whether six RGB indices are suitable for identifying invasive plant species in the QGIS environment on UAV images. To examine this, we determined the shoot area and number of common milkweed (*Asclepias syriaca*) and the inflorescence area and number of blanket flowers (*Gaillardia pulchella*) as two typical invasive species in open sandy grasslands. According to the results, the cover area of common milkweed was best identified with the TGI and SSI indices. The producers' accuracy was 76.38% (TGI) and 67.02% (SSI), while the user's accuracy was 75.42% (TGI) and 75.12% (SSI), respectively. For the cover area of blanket flower, the IF index proved to be the most suitable index. In spite of this, it gave a low producer's accuracy of 43.74% and user's accuracy of 51.4%. The used methods were not suitable for the determination of milkweed shoot and the blanket flower inflorescence number, due to significant overestimation. With the methods presented here, the data of large populations of invasive species can be processed in a simple, fast, and cost-effective manner, which can ensure the precise planning of treatments for nature conservation practitioners.

**Keywords:** biological invasions; blanket flower; common milkweed; drone (UAV); image processing; QGIS; remote sensing; RGB colour-based vegetation indices; spectral discrimination

## 1. Introduction

Invasive alien species (IAS) are species that usually enter a new habitat with human help (intentionally or accidentally), where their population size increases in a monotonous way in space and time. Alien species are one of the most serious causes of the degradation of the remaining natural habitats and the decrease of native species [1]. They severely damage ecosystem services, which would be essential for human well-being and survival, thus causing not only economic but also human health problems worldwide [2,3]. The damage caused by IAS and their treatment is growing exponentially; for example, in the EU alone it cost € 116.24 billion in 2020, compared to a loss of € 19.64 billion in 2013 [4].

In protected areas, the management and control of IAS is one of the most important tasks of nature conservation [5]. Therefore, it is important to know the current distribution and population dynamics of IAS, which is only possible with proper monitoring methods [6]. However, accurate survey of IAS over larger areas (e.g., hectares) by using

traditional methods (based on transects or quadrats) is extremely limited in terms of time, energy, and cost. The disturbance of vegetation in the traditional survey (e.g., trampling, soil compaction, etc.) may cause further problems [7,8], which may create favourable conditions for the further establishment of invasive plant species [9]. Remote sensing (RS) has long been preferred for mapping invasive species (mainly invasive plant species) because of its ability to provide synoptic images of large geographic areas. This is an advantage over field surveys, which are often limited to small areas and almost impossible in hard-to-reach places. Historically, RS has been crucial in the identification of IAS [7,8]. Satellite RS can provide important information about the general characteristics of soil forms and vegetation types, but often the spatial and temporal resolution is not good enough to determine the distribution of each species or to fine tune the landscape characteristics between vegetation types. In addition, the available satellite images are not always acquired during the desired phenological stage for a given species or vegetation type [7,8,10]. Therefore, they usually are more synopsis in nature. One possible solution to this problem could be to develop a new RS-based method with unmanned aerial vehicles (UAVs), or more commonly called drones. There are several advantages of their application described in the literature. Monitoring can be performed without disturbing the vegetation. Drone-based surveys may cover a much larger area, but it should be noted that the size of the surveyed vegetation stands (few hectares) is between the size of traditional field surveys and satellite-based surveys. Their resolution is suitable not only for vegetation, but sometimes even for a finer identification of species. Many authors point out that time and costs of work are also significantly reduced [11–14]. Taking advantage of these benefits, drones have been used successfully to study many IAS, in a variety of ecosystems, including grassland habitats [15,16].

Grasslands have significant biodiversity, carbon sequestration, and their primary production and habitats are also vulnerable to damage by IAS [8]. However, in the case of grasslands, the observation of these species, or even their recognition at all, can cause serious difficulties with RS. They may show spectral properties similar to native species or grow with native species, so their visual separation from the "background" can be quite challenging. In response to these constraints, monitoring of invasive species by RS is often only possible indirectly. Indirect methods often rely on data and models from multiple sources to identify IAS and thereby estimate their distribution [8]. Phenological divergence (e.g., flowering, dormancy phenology), for example, may help to distinguish between native vegetation and invasive species [7,8,17,18].

For the EU, the Pannonian sand grasslands in Hungary are important habitats that are threatened by the spread of many IAS [19,20]. In the present study, we examined two of them in the open sandy grassland habitats of Lake Kolon: common milkweed (*Asclepias syriaca* L.) as one of the most common and dangerous transformer species. The other is blanket flower (*Gaillardia pulchella* Foug.), which shows a remarkable population increase in the studied area, and will probably cause serious problems for nature conservation in the future. The characteristic colours and shapes of the two species are easily identifiable, and they are easily distinguishable from other species at higher resolutions, which always makes it easier to distinguish them. Thus, plant number and cover estimates for the two target species were determined based on RGB images.

The use of UAV-based remote sensing for identifying invasive plant species allows for the generation of higher spatial resolution aerial data of specific areas compared to unmanned aerial platforms or satellite sensors [21–24]. Various sensors have been used to monitor invasive species on different platforms over the last 15–20 years [25]. The methods used to classify invasive vegetation in scientific literature are diverse and of varying complexity. Among others, we encounter spectral angle mapping (SAM) [26–28], support vector machines (SVMs) [29,30], regression trees [29], and MaxEnt [29–32]. Several studies have applied SVMs and ANN to map the distribution of milkweed.

To analyse RGB images, three simple and three structured indices were used: R-G, R-B, G-B (Simple Difference Indices) [33], and Triangular Greenness Index (TGI) [34], Shape Index (IF) [35] and Spectral Shape Index (SSI) [36].

Many methods exist to classify high-resolution RGB images. In this study, we present a novel method that is transparent, robust, and does not require additional input features. The method does not require prerequisites about the statistical distribution of the input data, like for many other traditional classification methods (e.g., maximum likelihood). Our method also allows for explanation of the results, which is sometimes difficult for classification results of neural networks, where the values of weights of the model cannot directly be used for interpretation. Furthermore, computing requirements are relatively small, compared to machine learning approaches like random forest, support vector machine (SVM), and especially ANN or deep learning methods. Finally, our method does not require extra functionality from the software packages that were used. All steps of the method can be conducted without external packages, libraries, or algorithms.

Control of invasive alien species reflects an approach to agricultural weed control, as its foundations have been laid with its help [37]. The study focuses on developing a monitoring procedure for two invasive plant species, rather than treating them. The aim is to provide a way to accurately identify and assess the invasive species in a semiautomatic way, which can be cost-effective and accurate. This approach can help with planning treatment and analysing its effectiveness. The study can also serve as a model for monitoring other invasive or protected plant species.

The aim of the study was to identify two invasive plant species on real-color drone images by using six RGB indices in the QGIS environment. With the applied method, the coverage of the target vegetation ($m^2$) and the number of individuals were determined (number). Two types of validation were used for this: polygon-based and pixel-based (using a confusion matrix).

## 2. Materials and Methods

### 2.1. Studied Species

#### 2.1.1. Common Milkweed—*Asclepias syriaca*

Common milkweed belongs to the milkweed family (Asclepiadaceae) and is a perennial. Its shoots are 80–150 cm with high, bright green leaves, and the leaves are in cross-opposite positions and they have wide leaf blades, so they can be well identified in the study area based on their characteristic shape and colour. Common milkweed is native to North America and was introduced to Europe in 1629. It is currently present in about 23 European countries, in many habitat types [38,39]. The species is considered as one of the most dangerous invasive species [40–42]. It prefers loose soils with good drainage, and therefore it mainly spreads on sandy soils. Its thick roots run parallel to the soil surface, at a depth of 10–40 cm, but can also penetrate into deeper soil layers (1–3.8 m), which ensure its vegetative reproduction [38]. The invasion of milkweed is the most dynamic in those plant communities that have already been degraded by some effect or disturbance [9,43]. It can impede the regeneration of seminatural vegetation [9,44]. Its efficient vegetative (clonal) growth makes it suitable for slow but hardy space occupancy, even in open sandy grasslands [9,43,44]. It has already transformed natural vegetation in significant areas, but it also threatens the remaining natural ones. Given that the extent of degraded areas is increasing, milkweed is occupying them at a similar rate [43,45,46].

#### 2.1.2. Indian Blanket Flower—*Gaillardia pulchella*

Indian blanket flower belongs to the sunflower family (Asteraceae). It is a short-lived perennial or annual plant, and it is native species to northern Mexico and the southern and central United States. It prefers arid, sunny habitats and sandy or well-drained soils [47]. In the inflorescence of *G. pulchella* the ray flowers are deeply lobed, the inside half of them is purple and the outsides are yellowish. They have an appendix on the inner involucral bract. The leaves are undivided and have intact edges. In Hungary, its flowering lasts

from July to October [48]. The species is used as an ornamental plant due to its decorative, strikingly coloured inflorescences, so it has been introduced to many countries around the world. The inflorescences of the blanket flower are well detectable and should not be confused with other native species in the study area. In Hungary, the behaviour of *G. pulchella* is invasive [49]. This is further reinforced by the fact that other species of the family of Asteraceae may most likely become invasive species too [50,51]. Surprisingly, the monitoring, spread, and invasion of the species in the area have been dealt with by quite a few studies. In view of this and the fact that the blanket flower is present abundantly in the area of Lake Kolon, its establishment may be a cause for nature conservation concern. For the identification of blanket flower, its inflorescence was used because they are purple, yellowish, and larger (up to 5 cm in diameter) than other species' flowers or inflorescence in this habitat.

### 2.2. Study Area

The study of the two invasive species was carried out in the Lake Kolon, which is part of the Kiskunság National Park in 2020. The protected area is located in the westernmost part of the sand dunes between the Danube and the Tisza, on 3058 hectare, which developed from a tributary of the Danube in the postglacial period [52,53]. The average depth of the lake is 1 m. It is in the late succession phase; therefore a significant part is covered with reeds. It is surrounded by marshes and grove forests, swampy and marshy meadows, with sand dunes in the western part. The lake is dominated by contiguous reeds in the north–south direction, interspersed with a few patches of open water which are maintained artificially. The area is rich in flora and fauna [52,54].

Two investigated sites (hereafter referred to as stands I. and II.) were designated in the sand dune areas (Figure 1) and surveyed on 6 July 2020. The habitat of the surveyed stands naturally regenerating old-fields, belongs to the endemic open sand grasslands or Pannonic steppes, which are of major importance to the European Union Habitat Directive (Natura 2000 code: 6260) [54]. Stand I. was 8.5 hectares while stand II. was 6.7 hectares (Figure 2). In both stands, the two invasive plants had significant cover. Open source QGIS software version 3.10 [55] was used to create Figure 1 showing the location of the stands.

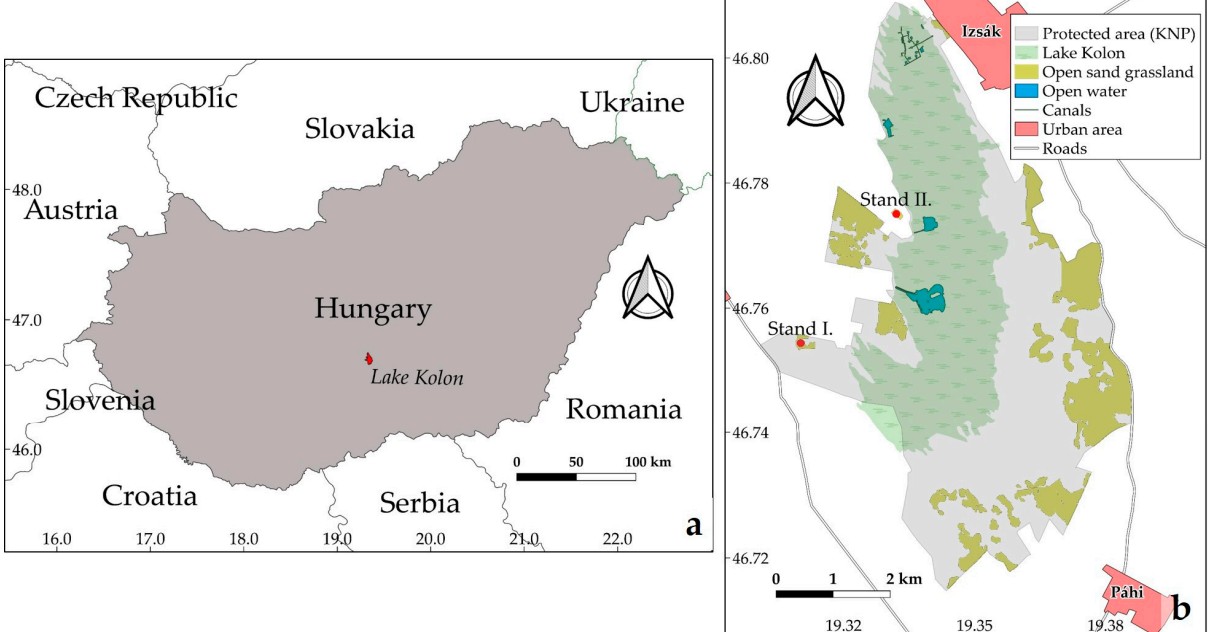

**Figure 1.** Location of the study site (3058 hectares) (**a**) in Hungary and (**b**) two selected stands connected with open sand grasslands.

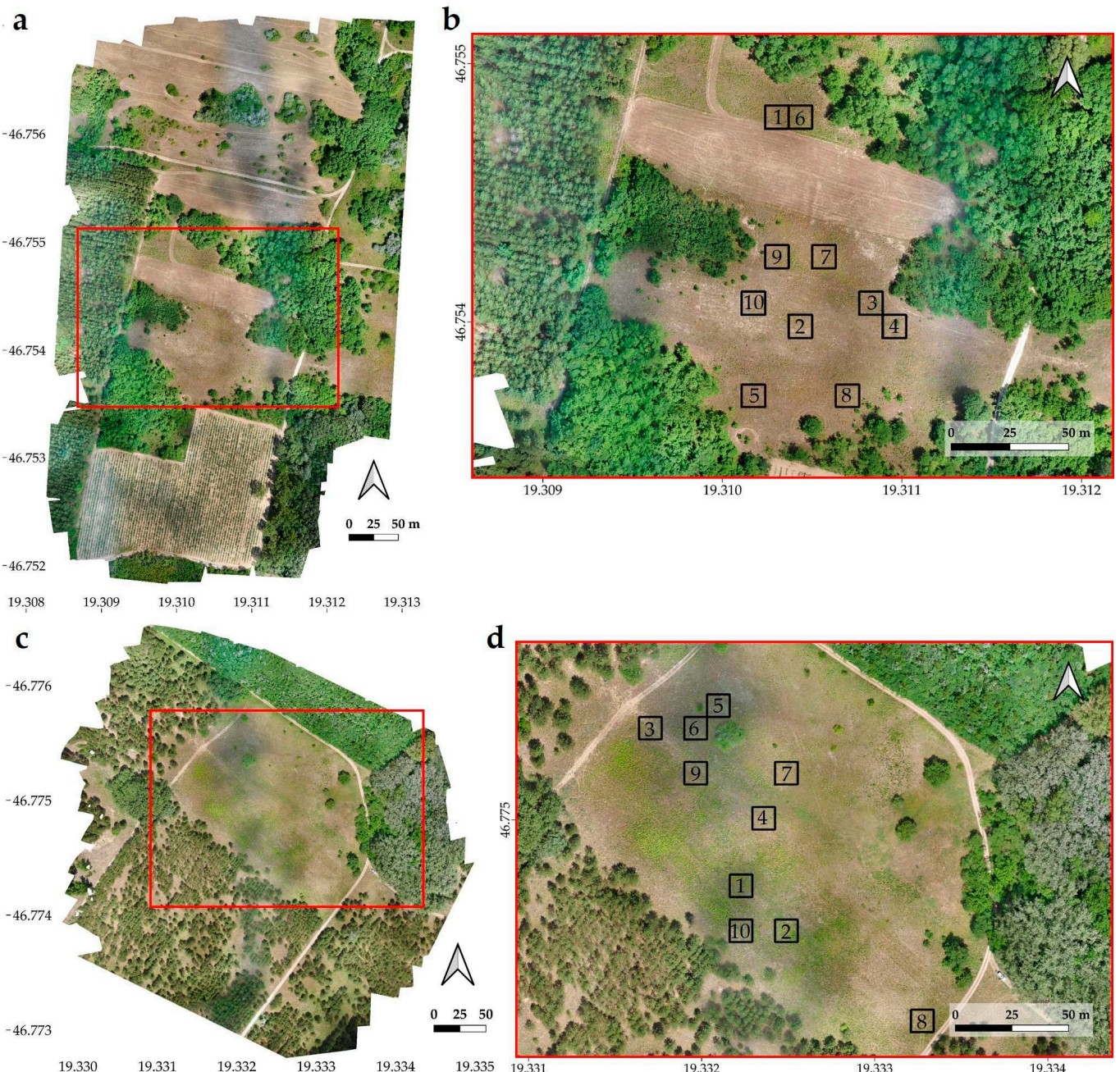

**Figure 2.** Ortomosaics of the stand I. (**a**) and II. (**c**). Locations of randomly selected 10 m × 10 m study areas (1–10) in stand I. (**b**) and in stand II. (**d**).

### 2.3. Documentation Methods

The two stands near Lake Kolon were flown with a senseFly eBee X fixed-wing drone owned by the Applied Geoinformatics and Remote Sensing Laboratory of the Department of Geoinformatics, Physical and Environmental Geography, University of Szeged. The drone wingspan is 116 cm, weighs 1.1–1.4 kg (depending on payload), and has a maximum flight time of 45 min. Its maximum radio connection distance is 3 km/1.9 mi [56]. During the drone flight, a senseFly S.O.D.A. was used as RGB camera. This camera has the following parameters: RTK/PPK support, 5472 × 3648 pixels resolution, shutter speed was global shutter 1/30–1/2000s, and JPEG image format [57].

A single flight was performed in both areas with the drone-mounted senseFly S.O.D.A. camera. The flights were executed according to preplanned flight routes. The average flight speed was 13 m/s. In order to have as many overlapping images as possible, and



thus more accurate data on the two sample areas, in both cases, we increased the length of the flight path, and thus the size of the photographed area as well. The applied flight parameters for the two stands were: flight altitude of 53.1 m/AED, flight time of 14 min and 51 s for stand I. and 16 min and 1 s for stand II. A total of 476 images were taken from stand I., while 360 images were taken from stand II. The images had 5 cm/pixel resolution and 60% longitudinal and 80% transverse image overlap. The weather conditions at the time of the flight as well as in the week ahead of the flight were sunny, dry, and free of precipitation and humidity.

*2.4. Data Processing Methods*

Agisoft Metashape Professional 1.5.4. photogrammetric software [58] was used to create orthophotos of the two stands from the aerial images. During the workflow in the Agisoft program, ground reference points were not used, instead of this the so-called onboard references (OtF meaning on_the_fly reference method) were used. The X, Y, and Z coordinates of the image center points were measured with an onboard GNSS with geodetic accuracy. Although the eBee UAV is an RTK capable platform, the corrections were applied during postprocessing.

In the orthophotos of both stands, 10 study areas were selected and each was 10 m × 10 m in size. These study areas were homogeneous patches with good representation of stands. For the selection of the study areas, the primary criterion was to ensure that the two invasive species were well represented and as homogeneously covered as possible, but it was also tried to select them randomly and in a spatially dispersed manner (Figure 2).

To identify invasive species in the study areas, the study used vegetation indices which are used in agriculture and/or forestry and applied bands of RGB images. The workflow took place in the QGIS [55] environment. The schematic illustration of this is shown in Figure 3.

2.4.1. The Used RGB Indices

In order to identify and select the two invasive species in the study areas, six RGB indices were calculated from drone images for the purpose of classification. Three of them are simple indices that can be calculated from the difference between red, green, and blue values [33]. Their equations are as follows:

$$R\text{-}G = R - G \tag{1}$$

$$R\text{-}B = R - B \tag{2}$$

$$G\text{-}B = G - B, \tag{3}$$

where R is the red band, G is the green band and B is the blue band. These simple difference indices show a significant relationship with photosynthetic pigments (chlorophyll-a and -b) contents; therefore they are often incorporated into other more structured vegetation indices (mentioned below). However, by themselves the indices may be suitable for spectral discrimination of IAS.

Three more complex indices that use the simple difference indices were evaluated as well. The TGI [34] estimates the chlorophyll concentration of leaves and foliage based on the area of a triangle formed by three characteristic reflectance points. TGI can be calculated using the formula

$$TGI = -0.5 \times [(\lambda r - \lambda b) \times (R - G) - (\lambda r - \lambda g) \times (R - B)], \tag{4}$$

where $\lambda r = 670$ nm, $\lambda g = 550$ nm, and $\lambda b = 480$ nm. The absorbance of these three wavelengths is determined by the amount of chlorophyll-a and -b. The three wavelengths define a triangle, the area of which is used to calculate the TGI. The area of the triangle is essentially given by the variation in the reflectance of the 550 nm wavelength. If the chlorophyll content decreases, the area of the triangle becomes larger, because the reflectance

at 550 nm increases strongly. Examination of PROSPECT leaf model simulations have shown that green reflectance increases when chlorophyll content decreases due to nitrogen deficiency [34,59,60]. Chlorophylls (and also carotenoids, which are dyes that increase the efficiency of photosynthesis) are strongly absorbed at 480 nm, so the reflectance at blue wavelengths does not change as the chlorophyll content decreases. Chlorophyll-a has a much higher absorption coefficient at 670 nm than at 550 nm, so as chlorophyll content decreases, the increase is greater at 550 nm than at 670 nm [34].

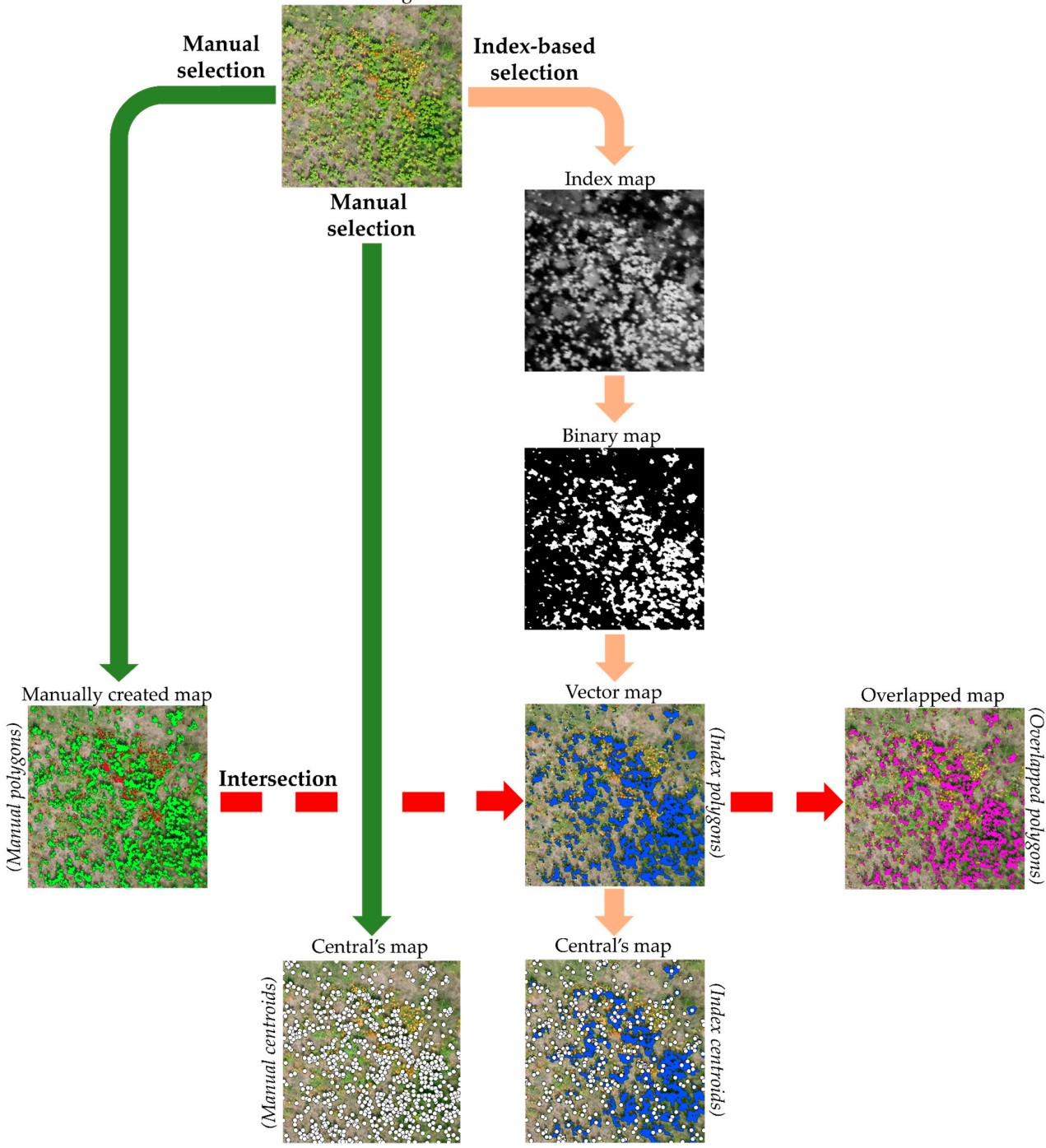

**Figure 3.** Schematic diagram of the workflow with study areas in QGIS. The two main selection routes (manual and index-based selections) and intersection for target species derived from RGB drone images. The study was based on the comparison of the results obtained from these.

The calculation of the IF index (shape index) is based on the formula [35]:

$$\text{IF} = \frac{2 \times \text{R} - \text{G} - \text{B}}{\text{G} - \text{B}} \tag{5}$$

IF describes the general shape of the spectral signature curve and allows the identification and distinction of soils and rock formations according to lithology [61].

The spectral shape index (SSI) allows good separation of shadows found in the vegetation [36]. Moreover, SSI showed a remarkable results for isolation of intense green leaves and blue and red shades of flowers [62]. The SSI is computed as

$$\text{SSI} = |\,\text{R} + \text{G} - 2 \times \text{G}\,|. \tag{6}$$

2.4.2. Data Validation Processes

The accuracy of RGB indices-based separation of the two target species was estimated by the following validation method. For this, we created polygons and centroids for the target vegetation of the study areas. The area of the polygons refers to the cover (in m$^2$), while the number of centroids indicates the number of shoots and inflorescences. The validation workflow can be divided into three parts.

First, the milkweed shoots and blanket flower inflorescences were manually delineated in each study area. This means that the shoots or shoot groups, inflorescences, or inflorescence groups were delineated by hand. On these delineated polygons (so-called, hereafter, manual polygons), the areas of the shoot and inflorescence were only indicated. Moreover, each individual shoot and inflorescence was marked manually too (hereafter, manual centroids) for the realistic indication of the shoot and inflorescence numbers (Figure 3).

Secondly, by using the applied indices, polygons and centroids were created for the two target species in QGIS. For this, the index-based selection was used to create so-called index polygons with indices. Six RGB indices were used to examine their applicability to separate the two invasive species compared to the manual delineation method. By using an RGB index, we created a so-called index map from a study area. Next, the index map was converted into a binary map by specifying a threshold value that best separated the target species. To create binary maps, thresholds must be defined for each study area by using the value ranges in the index maps (Table S1). The threshold values were determined based on the comparison of manually delineated polygons with index polygons created by using index-based selection in QGIS. Six RGB indices were used to examine their applicability in separating the two invasive species, and the value ranges of each index were analyzed to determine the characteristic intervals and sensitivity to the target species. For each study area, threshold values were defined to create binary maps from the index maps that most closely matched the areas of the manual polygons. Overall, the threshold values were selected based on the value ranges of each index and the desired accuracy of the index polygons compared to the manual polygons. The essence of applying a threshold is to reclassify pixels below it to 0 and above it to 1, thus obtaining a binary map. Then, the resulting binary map was vectorized. The vector map contains the index polygons (Figure 3). To illustrate the number of index polygons, their centroids were created (hereafter index centroids) (Figure 3). To check the number of shoots and inflorescences of the two target species, the number of index centroids were compared to the number of manual centroids.

Thirdly, the area and location of the index polygons were compared with the manual polygons. For this, intersection was used from the index polygons and manual polygons, and thus overlapping polygons are obtained from it. These overlapping polygons show the accuracy (area and location) available with index polygons (Figure 3). If the three polygon types are displayed together, manual polygons can be perceived as false negatives, index polygons as false positives, and overlapped polygons as true positive areas. The explanation for this is that where the manual polygons are visible, the index did not find the areas that belong to the target vegetation. Thus, there was no overlap between index

polygons and manual polygons. The index polygons are visible if the area delineated by the index does not belong to the target vegetation. This is because there is no intersection with the manual polygons. In the case of overlapped polygons, there is an intersection between the manual polygons and the index polygons. These areas are completely part of the target vegetation. Figure 4 illustrates the connection between the three polygon types.

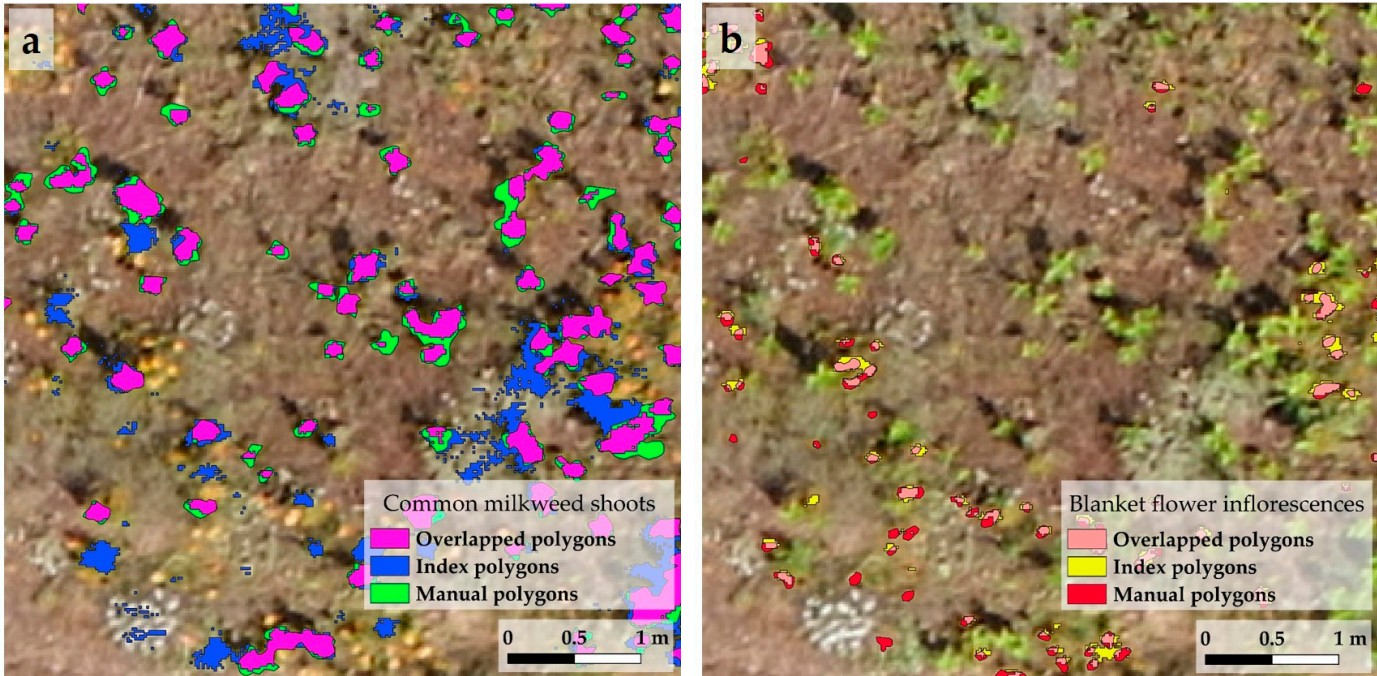

**Figure 4.** Overlapping polygons were used to check the accuracy of the area and locations of the index polygons. These overlapping polygons are derived from the intersection of manual polygons and index polygons. In the case of milkweed (**a**), the green areas indicate the false negative, blue are the false positive, and purple polygons refer to the true positive areas. (**b**) The same maps were made for blanket flower, where the red areas are the false negatives, yellow are the false positive areas, and pink areas indicate the true positive ones. The image used for illustration shows a detail of the first study area of the first stand. Moreover, the index polygons were formed with the G-B index in the case of milkweed, and the IF index in the case of blanket flower.

Accuracy assessment can be used to evaluate the classification results of remotely sensed data. The confusion matrix with the reference and classified pixels can be used to determine the overall accuracy, the Kappa coefficient, and the producer's and user's accuracy values for the classes [62]. The overall accuracy means the ratio between the reference pixels correctly classified by the classification algorithm and all reference pixels. In this case, this is the ratio of the pixels of rasterized manual polygons and binary maps. The producer's accuracy value shows the ratio of the number of pixels correctly classified by index for a given plant to the number of pixels delimited by a manual polygon for the given plant. This is closely related to the omission error, which shows the percentage of pixels in the reference category that were placed in the wrong class (false positive). The value of the user's accuracy means the ratio of the number of pixels correctly classified by index for a given plant to the total number of pixels classified by index for the given plant. The commission error shows the percentage of elements classified in a given class that were wrongly classified (false negative). In general, it is better to consider the Kappa values, and the producer's and user's accuracy instead of the overall accuracy, because the latter can be misleading due to the many correctly classified background pixels (Tables S4 and S5).

*2.5. Statistical Analysis*

To check whether the applied threshold of the indices show a dependence between the two stands, an unpaired (two tailed) *t*-test was used. Correlation (Pearson r) and simple percentage comparison were used to determine the suitability of the RGB indices. Statistical analyses were performed in Graph Pad Prism 8.0.1.244 for Windows (GraphPad Software, La Jolla, CA, USA). The lowest significant value was $p = 0.05$, higher $p$ values are not considered significant.

### 3. Results

To identify the two invasive plant species on drone images in QGIS, six RGB indices were used. The present work examined the characteristic intervals of each index, the sensitivity of the index to the target species (which index can be applied to which species, and which threshold values are best used to select them). Threshold values were applied such that the delineated areas (m$^2$) were the most similar to the areas (m$^2$) of the manual polygons in the given study area, and a given illumination spectrum and soil and plant surface moisture, respectively.

In the case of simple difference indices, the index intervals and the used threshold values were as follows. The intervals of the R-G index were between −59 and 82 (Table S1). This index was applicable to both target species. In the index map, milkweed shoots appeared as black spots, while blanket flower inflorescences were white (Figure 5). In order to create binary maps from index maps, threshold values between −23 and 5 were applied in the case of milkweed shoots, while in the case of blanket flower, the used threshold values were between 21 and 42 (Table 1). The intervals of the R-B index map were between −12 and 179 (Table S1). On these index maps, the inflorescences of the blanket flower can be easily selected, as they appeared as black spots (unlike on the previous map, where they were white ones). The used threshold values for creating binary maps were between 72 and 108 (Table 1). The intervals were between −8 and 141 on the G-B index map (Table S1). In the G-B index maps, the milkweed shoots appeared as black spots, which made it possible to sort them. For the formation of the binary maps, we used 52 to 79 threshold values in the study areas (Table 1).

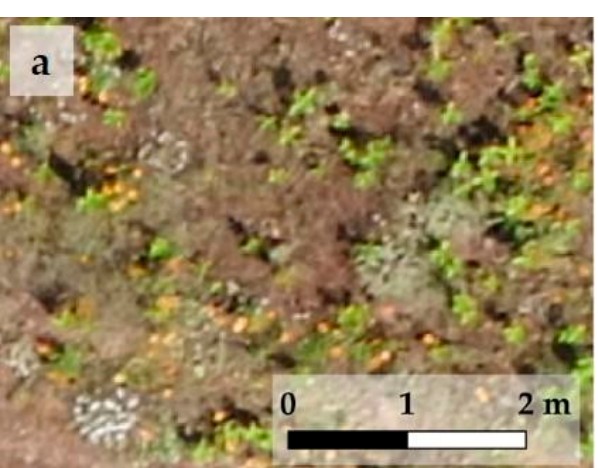 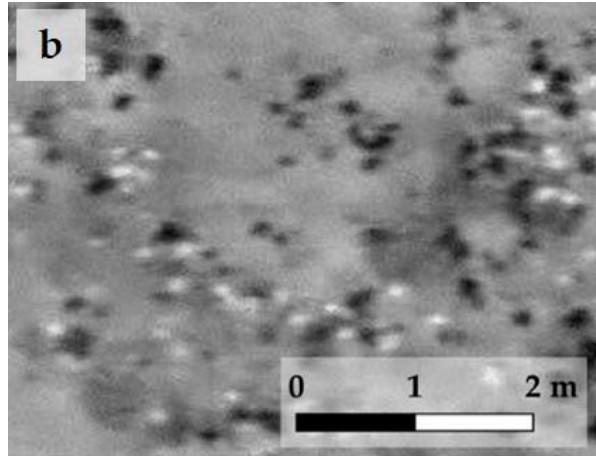

**Figure 5.** A detail of the RGB image of the first study area of the first stand before R-G index selection (**a**) and after (**b**). In the RGB image (**a**), the bright green milkweed shoots, and they are clearly recognizable as black spots in the index map (**b**), while the blanket flower inflorescences are orange, purple in RGB image (**a**) and appear as white spots in the index map (**b**).

**Table 1.** The identified threshold values of the indices for identification of common milkweed shoots and blanket flower inflorescences.

| Applicability to Species | | Milkweed | Blanket Flower | Blanket Flower | Milkweed | Milkweed | Blanket Flower | Milkweed |
|---|---|---|---|---|---|---|---|---|
| Applied Thresholds | | R-G | R-G | R-B | G-B | TGI | IF | SSI |
| STAND I. | 1.study area | <−1 | >38 | >99 | >71 | >4050 | >130 | >67 |
| | 2.study area | <5 | >37 | >100 | >64 | >3500 | >119 | >55 |
| | 3.study area | <−3 | >40 | >102 | >71 | >4250 | >135 | >72 |
| | 4.study area | <−5 | >35 | >87 | >68 | >4250 | >113 | >74 |
| | 5.study area | <−5 | >42 | >98 | >72 | >4150 | >135 | >70 |
| | 6.study area | <−1 | >38 | >108 | >70 | >4050 | >137 | >67 |
| | 7.study area | <−3 | >37 | >96 | >79 | >4250 | >129 | >69 |
| | 8.study area | <−3 | >38 | >92 | >71 | >4200 | >119 | >71 |
| | 9.study area | <−1 | >38 | >98 | >71 | >4050 | >128 | >68 |
| | 10.study area | <2 | >34 | >89 | >64 | >3550 | >111 | >58 |
| STAND II. | 1.study area | <−17 | >27 | >97 | >68 | >4500 | >116 | >82 |
| | 2.study area | <−22 | >25 | >98 | >78 | >5300 | >116 | >97 |
| | 3.study area | <−5 | >29 | >93 | >57 | >3500 | >110 | >60 |
| | 4.study area | <−7 | >30 | >94 | >72 | >4350 | >116 | >75 |
| | 5.study area | <−23 | >21 | >72 | >53 | >3500 | >88 | >65 |
| | 6.study area | <−17 | >23 | >78 | >52 | >3500 | >92 | >65 |
| | 7.study area | <−10 | >33 | >89 | >73 | >4550 | >116 | >79 |
| | 8.study area | <−11 | >38 | >101 | >73 | >4600 | >132 | >81 |
| | 9.study area | <−12 | >29 | >89 | >60 | >3850 | >110 | >68 |
| | 10.study area | <−12 | >24 | >86 | >57 | >3800 | >92 | >68 |

The index intervals and the used threshold values were as follows on the structured indices. The minimum and maximum intervals of the TGI index of the study areas ranged from −1080 to 9740 (Table S1). The TGI index was suitable for determining the milkweed shoots, because they were the white spots in the index map of the open sand grassland vegetation. For the formation of binary maps, the applicable threshold values were found to be 3500 and 4600 (Table 1). For IF, the minimum and maximum intervals of study areas were −37.656 and 260.651 (Table S1), and the applied thresholds were bigger than 88 and 137. This index was applicable to the delineation of blanket flower inflorescence. It shows the presence of inflorescences in the vegetation as white spots (Table 1) similarly to the R-G index (Figure 5). The intervals of the SSI index were between 0 and 179 (Table S1). The index was suitable for the separation of milkweed shoots in open sand grassland vegetation, as the shoots of the species appeared as black spots on the index maps. Binary maps can be created from the SSI index maps using 55 to 97 threshold values (Table 1).

To check whether the indices show a dependence between the two stands, a *t*-test was used. There were significant differences in the applied thresholds of R-G and IF between the two stands (Table 2). In the case of the R-G index, the threshold values suitable for both milkweed shoots and blanket flower inflorescences showed a strong significant difference between the two stands. Due to the intense green colour of the milkweed shoots and the bright purple and yellowish shades of the blanket flower inflorescences, the R-G values calculated from the RGB intensity showed lower values in areas covered with milkweed and higher values in the areas where the blanket flower dominated.

A threshold value was set that shows close to 100% similarity between the area of the index polygons and the area of the manual polygons in the study area for the given species. For both species, the greatest similarity could be achieved with the more structured indices (Tables S2 and S3). This similarity was also checked with correlation tests, according to which there is a strong significant similarity (Figure S1). However, this does not yet show how misleading the indices used for delineation were (false positive, false negative areas). This can be examined with overlapping polygons, as they not only show the areal similarity of manual polygons and index polygons, but also the matching of

their locations. In other words, they indicate how accurately the index polygons fall on the manual polygons (namely the target vegetation). Therefore, manual polygons were compared with overlapping polygons to check the coverage area and location. In the case of milkweed, the largest areal overlap can be achieved with the TGI index, it was 69.13%. The SSI and G-B indices show a percentage similar to the former, 69.05% and 65.34%. The smallest areal overlap was shown by the R-G index, this was 57.51% overlapping with the manual polygons (Table 3). Based on the correlation test, it can be concluded that manual polygons and polygons created with indices significantly correspond to each other ($p < 0.0001$) (Figure 6). Overall, it can be said that the indices calculated for common milkweed give a good indication of the cover of the species and match their location with great accuracy.

**Table 2.** Examination of the differences in the applied threshold values between the two stands. Unpaired *t* test (two-tailed) was used, n = 10. Significance was applied from a value of $p = 0.05$, values higher than this were not significant.

| Unpaired *t* Test | Stand I. vs. Stand II. | | | | | | |
|---|---|---|---|---|---|---|---|
| | R-G (Milkweed) | R-G (Blanket Flower) | R-B (Blanket Flower) | G-B (Milkweed) | TGI (Milkweed) | IF (Blanket Flower) | SSI (Milkweed) |
| *p* value | <0.0001 | <0.0001 | 0.0537 | 0.096 | 0.5934 | 0.0054 | 0.1002 |

**Table 3.** The common milkweed shoot area in the study areas. The percentage values showed the similarities of the overlapped polygons to the manual polygons. SD, standard deviation, n = 20.

| Milkweed SHOOT AREA | | Manual Polygons m$^2$ | R-G Overlapped Polygons | | G-B Overlapped Polygons | | TGI Overlapped Polygons | | SSI Overlapped Polygons | |
|---|---|---|---|---|---|---|---|---|---|---|
| | | | m$^2$ | % | m$^2$ | % | m$^2$ | % | m$^2$ | % |
| STAND I. | 1.study area | 8.601 | 5.235 | 60.86 | 5.719 | 66.49 | 6.077 | 70.65 | 6.139 | 71.37 |
| | 2.study area | 7.594 | 4.109 | 54.11 | 5.495 | 72.35 | 5.530 | 72.82 | 5.424 | 71.42 |
| | 3.study area | 8.440 | 4.496 | 53.23 | 4.026 | 47.66 | 5.362 | 63.48 | 5.341 | 63.24 |
| | 4.study area | 10.098 | 5.259 | 52.08 | 6.078 | 60.19 | 6.108 | 60.48 | 5.986 | 59.28 |
| | 5.study area | 1.984 | 1.049 | 52.87 | 1.082 | 54.54 | 1.232 | 62.11 | 1.247 | 62.86 |
| | 6.study area | 16.944 | 11.450 | 67.57 | 12.415 | 73.26 | 12.991 | 76.66 | 13.028 | 76.88 |
| | 7.study area | 2.393 | 1.113 | 46.52 | 1.022 | 42.72 | 1.361 | 56.86 | 1.423 | 59.44 |
| | 8.study area | 7.596 | 4.517 | 59.47 | 4.909 | 64.63 | 5.126 | 67.49 | 5.138 | 67.64 |
| | 9.study area | 9.456 | 6.046 | 63.93 | 6.482 | 68.55 | 6.894 | 72.90 | 6.849 | 72.42 |
| | 10.study area | 4.232 | 2.305 | 54.48 | 2.672 | 63.16 | 2.774 | 65.56 | 2.762 | 65.27 |
| STAND II. | 1.study area | 22.422 | 16.198 | 72.24 | 16.991 | 75.77 | 17.572 | 78.37 | 17.593 | 78.46 |
| | 2.study area | 24.573 | 16.550 | 67.35 | 19.116 | 77.79 | 19.210 | 78.17 | 19.101 | 77.72 |
| | 3.study area | 11.129 | 6.916 | 62.14 | 8.034 | 72.19 | 8.110 | 72.87 | 8.102 | 72.80 |
| | 4.study area | 4.222 | 2.140 | 50.68 | 2.409 | 57.06 | 2.584 | 61.20 | 2.580 | 61.11 |
| | 5.study area | 3.200 | 1.002 | 31.33 | 1.717 | 53.65 | 1.980 | 61.88 | 1.994 | 62.30 |
| | 6.study area | 5.904 | 3.279 | 55.55 | 4.087 | 69.23 | 4.215 | 71.40 | 4.163 | 70.51 |
| | 7.study area | 3.389 | 1.649 | 48.66 | 2.138 | 63.09 | 2.196 | 64.79 | 2.184 | 64.44 |
| | 8.study area | 8.088 | 3.631 | 44.89 | 5.331 | 65.91 | 5.147 | 63.64 | 4.960 | 61.32 |
| | 9.study area | 13.992 | 10.224 | 73.07 | 10.324 | 73.78 | 10.779 | 77.03 | 10.961 | 78.33 |
| | 10.study area | 20.889 | 16.531 | 79.13 | 17.730 | 84.88 | 17.588 | 84.20 | 17.582 | 84.17 |
| Average % | | | | 57.51 | | 65.34 | | 69.13 | | 69.05 |
| SD % | | | | 11.20 | | 10.47 | | 7.42 | | 7.41 |

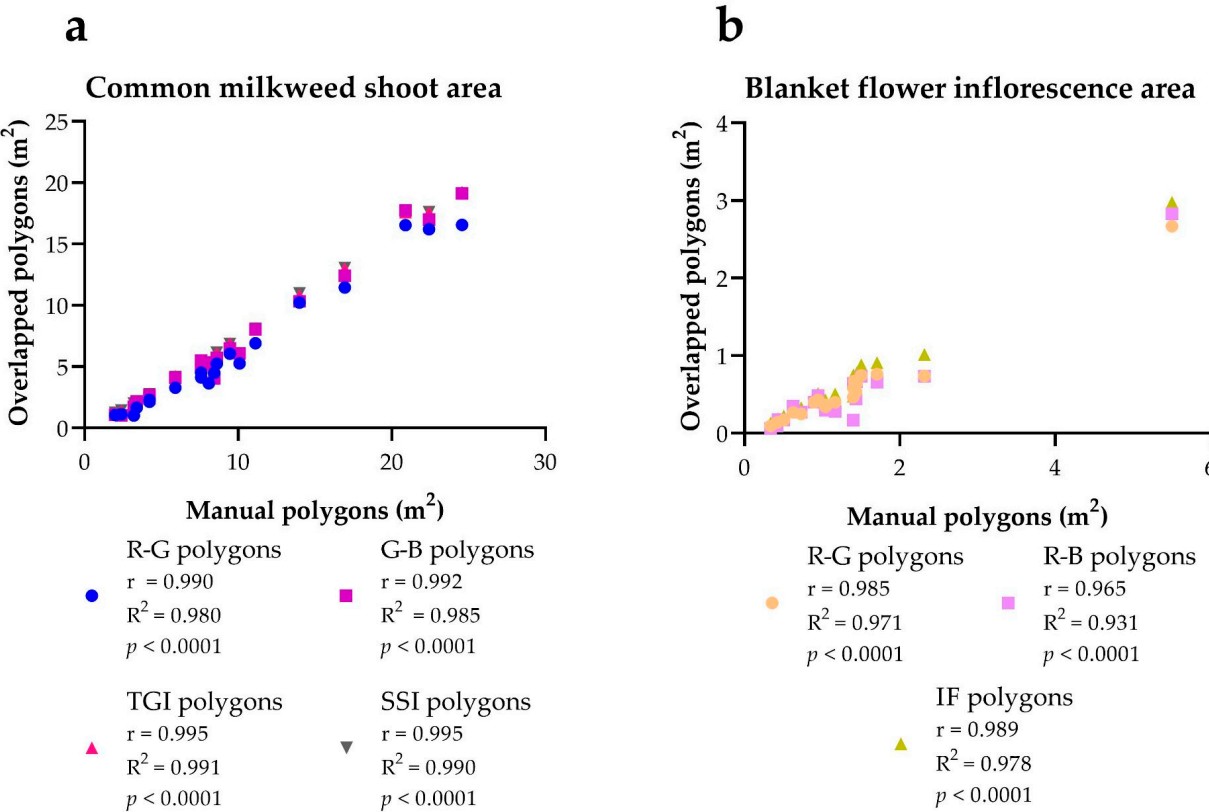

**Figure 6.** The correlation between the manual polygons and overlapped polygons (overlapping of the index polygons and manual polygons) in the case of the common milkweed (**a**) and blanket flower (**b**). Pearson r correlation (two-tailed) was used, n = 20. Significance was applied from a value of $p = 0.05$, values higher than this were not significant.

The areal overlap of the indices which used to separate the inflorescences of the blanket flower with the manual polygons showed a significantly lower percentage of similarities than in the case of the milkweed. However, here too, the more structured index (IF) showed the greatest overlap, 46.99%. The R-G and the R-B indices display lower similarities, 38.47% and 36.83%, respectively (Table 4). In contrast, a strong correlation was detected between the overlapping polygons and the manual polygons ($p < 0.0001$) (Figure 6). Thus, with the indices used to separate the inflorescences of the blanket flower, high accuracy can be achieved in terms of their location, but they are less informative for the area of the inflorescences.

**Table 4.** The blanket flower inflorescence area in the study areas. The percentage values showed the similarities of the overlapped polygons to the manual polygons. SD, standard deviation, n = 20.

| Blanket Flower Inflorescence Area | | Manual Polygons m² | R-G overlapped Polygons | | R-B Overlapped Polygons | | IF Overlapped Polygons | |
|---|---|---|---|---|---|---|---|---|
| | | | m² | % | m² | % | m² | % |
| STAND I. | 1.study area | 1.406 | 0.580 | 41.25 | 0.641 | 45.58 | 0.753 | 53.54 |
| | 2.study area | 1.402 | 0.464 | 33.11 | 0.168 | 12.01 | 0.479 | 34.22 |
| | 3.study area | 0.899 | 0.397 | 44.16 | 0.402 | 44.77 | 0.476 | 52.91 |
| | 4.study area | 1.439 | 0.533 | 37.03 | 0.442 | 30.74 | 0.677 | 47.08 |
| | 5.study area | 0.949 | 0.427 | 45.06 | 0.485 | 51.12 | 0.512 | 53.92 |
| | 6.study area | 1.445 | 0.64 | 44.27 | 0.664 | 45.92 | 0.774 | 53.56 |
| | 7.study area | 5.505 | 2.667 | 48.44 | 2.827 | 51.37 | 2.974 | 54.02 |
| | 8.study area | 2.316 | 0.737 | 31.83 | 0.735 | 31.75 | 1.015 | 43.81 |
| | 9.study area | 1.706 | 0.761 | 44.59 | 0.657 | 38.52 | 0.908 | 53.22 |
| | 10.study area | 1.169 | 0.398 | 34.09 | 0.280 | 23.95 | 0.504 | 43.12 |

**Table 4.** *Cont.*

| Blanket Flower Inflorescence Area | | Manual Polygons m² | R-G overlapped Polygons | | R-B Overlapped Polygons | | IF Overlapped Polygons | |
|---|---|---|---|---|---|---|---|---|
| | | | m² | % | m² | % | m² | % |
| STAND II. | 1.study area | 1.506 | 0.746 | 49.55 | 0.734 | 48.78 | 0.881 | 58.54 |
| | 2.study area | 1.413 | 0.67 | 47.43 | 0.639 | 45.28 | 0.747 | 52.85 |
| | 3.study area | 0.421 | 0.137 | 32.66 | 0.093 | 22.16 | 0.156 | 37.09 |
| | 4.study area | 1.043 | 0.365 | 35.05 | 0.339 | 32.52 | 0.431 | 41.33 |
| | 5.study area | 0.629 | 0.268 | 42.58 | 0.347 | 55.15 | 0.347 | 55.11 |
| | 6.study area | 0.509 | 0.172 | 33.75 | 0.172 | 33.86 | 0.221 | 43.34 |
| | 7.study area | 1.046 | 0.34 | 32.50 | 0.295 | 28.22 | 0.368 | 35.20 |
| | 8.study area | 0.436 | 0.129 | 29.62 | 0.174 | 39.94 | 0.187 | 42.96 |
| | 9.study area | 0.731 | 0.248 | 33.94 | 0.270 | 36.94 | 0.327 | 44.78 |
| | 10.study area | 0.336 | 0.096 | 28.53 | 0.060 | 18.05 | 0.132 | 39.18 |
| Average % | | | | 38.47 | | 36.83 | | 46.99 |
| SD % | | | | 6.76 | | 11.95 | | 7.41 |

The individual numbers were also checked by comparing manual centroids and index centroids.

The number of centroids given by R-G correlated with the number of manual centroids at the *p* = 0.01 level for the shoot numbers of milkweed. The number of centroids obtained with the other indices (G-B, TGI, and ISS) did not show a significant correlation. The number of centroids of the indices used for the selection of blanket flower inflorescences did not correlate with the number of manual centroids in any case (Figure 7).

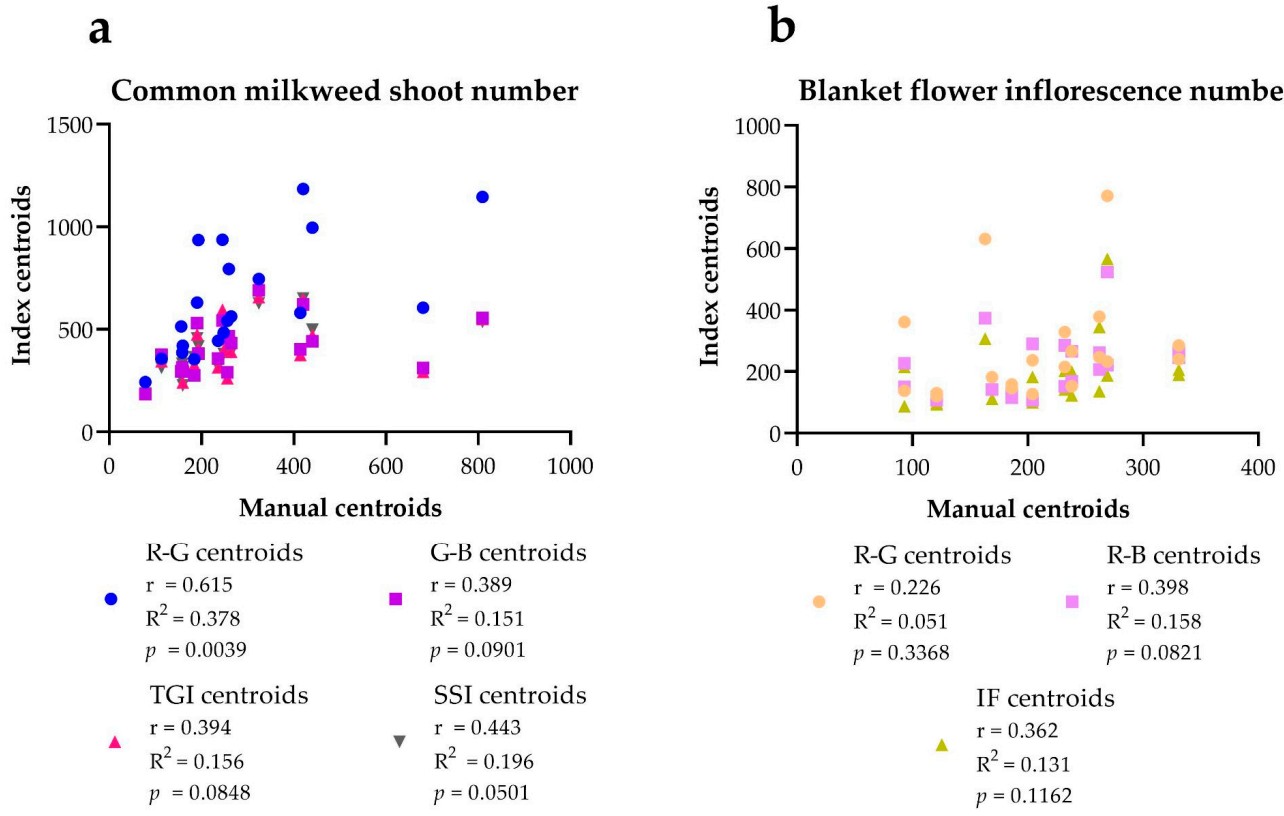

**Figure 7.** The correlation between the manual centroids (manually sorted centroids) and overlapped centroids (overlapping of the index centroids and manual centroids) in the case of the common milkweed (**a**) and blanket flower (**b**). Pearson r correlation (two-tailed) was used, n = 20. Significance was applied from a value of *p* = 0.05, values higher than this were not significant.

The overestimation of the milkweed shoot number and the blanket flower inflorescences number are typical for the index centroids numbers (Tables 5 and 6). The index-based delineation, contrary to our expectation, achieved a more fragmented delineation, which is why the index centroids numbers were higher. This was best experienced in the case of milkweed shoots, which is illustrated in Figure 8. An exception was the IF index, which was used to separate blanket flower inflorescences, which shows a 93.87% similarity with the manual centroids, but this was only achieved with a high standard deviation (Table 6).

**Table 5.** The common milkweed shoot numbers in the study areas. The percentage values showed the similarities of the index centroid (R-G, G-B, TGI and SSI centroids) numbers to the manual centroids numbers. SD, standard deviation, n = 20.

| Milkweed Shoot Number | | Manual Centroids Number | R-G Centroids | | G-B Centroids | | TGI Centroids | | SSI Centroids | |
|---|---|---|---|---|---|---|---|---|---|---|
| | | | Number | % | Number | % | Number | % | Number | % |
| STAND I. | 1.study area | 259 | 794 | 306.56 | 467 | 180.3 | 437 | 168.72 | 414 | 159.84 |
| | 2.study area | 193 | 936 | 484.97 | 382 | 197.92 | 404 | 209.32 | 417 | 216.06 |
| | 3.study area | 236 | 444 | 188.13 | 358 | 151.69 | 315 | 133.47 | 315 | 133.47 |
| | 4.study area | 248 | 484 | 195.16 | 458 | 184.67 | 399 | 160.88 | 380 | 153.22 |
| | 5.study area | 78 | 243 | 311.53 | 184 | 235.89 | 204 | 261.53 | 188 | 241.02 |
| | 6.study area | 420 | 1184 | 281.9 | 622 | 148.09 | 646 | 153.8 | 650 | 154.76 |
| | 7.study area | 113 | 357 | 315.92 | 377 | 333.62 | 344 | 304.42 | 313 | 276.99 |
| | 8.study area | 190 | 630 | 331.57 | 531 | 279.47 | 475 | 250 | 455 | 239.47 |
| | 9.study area | 264 | 563 | 213.25 | 433 | 164.01 | 390 | 147.72 | 388 | 146.96 |
| | 10.study area | 156 | 514 | 329.48 | 296 | 189.74 | 298 | 191.02 | 264 | 169.23 |
| STAND II. | 1.study area | 245 | 937 | 382.44 | 543 | 221.63 | 596 | 243.26 | 532 | 217.14 |
| | 2.study area | 809 | 1145 | 141.53 | 555 | 68.6 | 551 | 68.1 | 537 | 66.37 |
| | 3.study area | 440 | 995 | 226.13 | 442 | 100.45 | 470 | 106.81 | 498 | 113.18 |
| | 4.study area | 159 | 420 | 264.15 | 294 | 184.9 | 240 | 150.94 | 227 | 142.76 |
| | 5.study area | 184 | 354 | 192.39 | 276 | 150 | 328 | 178.26 | 364 | 197.82 |
| | 6.study area | 256 | 542 | 211.71 | 290 | 113.28 | 262 | 102.34 | 281 | 109.76 |
| | 7.study area | 158 | 387 | 244.93 | 316 | 200 | 310 | 196.2 | 332 | 210.12 |
| | 8.study area | 324 | 746 | 230.24 | 690 | 212.96 | 657 | 202.77 | 627 | 193.51 |
| | 9.study area | 414 | 580 | 140.09 | 403 | 97.34 | 375 | 90.57 | 373 | 90.09 |
| | 10.study area | 680 | 606 | 89.11 | 312 | 45.88 | 293 | 43.08 | 304 | 44.7 |
| Average % | | | | 254.06 | | 173.02 | | 168.16 | | 163.82 |
| SD % | | | | 92.23 | | 68.94 | | 67.22 | | 61.01 |

**Table 6.** The blanket flower inflorescence numbers in the study areas. The percentage values are derived from the index centroid (R-G, R-B and IF centroids) numbers to the manual centroids numbers. SD, standard deviation, n = 20.

| Blanket Flower Inflorescence Number | | Manual Centroids Number | R-G Centroids | | R-B Centroids | | IF Centroids | |
|---|---|---|---|---|---|---|---|---|
| | | | Number | % | Number | % | Number | % |
| STAND I. | 1.study area | 331 | 285 | 86.10 | 245 | 74.01 | 190 | 57.40 |
| | 2.study area | 262 | 379 | 144.65 | 262 | 100.00 | 345 | 131.67 |
| | 3.study area | 121 | 130 | 107.43 | 107 | 88.42 | 94 | 77.68 |
| | 4.study area | 232 | 329 | 141.81 | 285 | 122.84 | 203 | 87.50 |
| | 5.study area | 186 | 159 | 85.48 | 131 | 70.43 | 125 | 67.20 |
| | 6.study area | 204 | 237 | 116.17 | 290 | 142.15 | 183 | 89.70 |
| | 7.study area | 269 | 771 | 286.61 | 524 | 194.79 | 566 | 210.40 |
| | 8.study area | 163 | 631 | 387.11 | 374 | 229.44 | 307 | 188.34 |
| | 9.study area | 238 | 266 | 111.76 | 266 | 111.76 | 200 | 84.03 |
| | 10.study area | 93 | 361 | 388.17 | 227 | 244.08 | 215 | 231.18 |

**Table 6.** *Cont.*

| Blanket Flower Inflorescence Number | | Manual Centroids Number | R-G Centroids | | R-B Centroids | | IF Centroids | |
|---|---|---|---|---|---|---|---|---|
| | | | Number | % | Number | % | Number | % |
| STAND II. | 1.study area | 331 | 240 | 72.50 | 262 | 79.15 | 206 | 62.23 |
| | 2.study area | 262 | 247 | 94.27 | 207 | 79.00 | 136 | 51.90 |
| | 3.study area | 121 | 120 | 99.17 | 118 | 97.52 | 98 | 80.99 |
| | 4.study area | 232 | 215 | 92.67 | 152 | 65.51 | 142 | 61.20 |
| | 5.study area | 186 | 146 | 78.49 | 115 | 61.82 | 120 | 64.51 |
| | 6.study area | 204 | 127 | 62.25 | 107 | 52.45 | 100 | 49.01 |
| | 7.study area | 269 | 233 | 86.61 | 221 | 82.15 | 188 | 69.88 |
| | 8.study area | 169 | 182 | 107.69 | 142 | 84.02 | 112 | 66.27 |
| | 9.study area | 238 | 153 | 64.28 | 169 | 71.00 | 123 | 51.68 |
| | 10.study area | 93 | 138 | 148.38 | 150 | 161.29 | 88 | 94.62 |
| Average % | | | | 138.08 | | 110.59 | | 93.87 |
| SD % | | | | 98.00 | | 55.88 | | 53.91 |

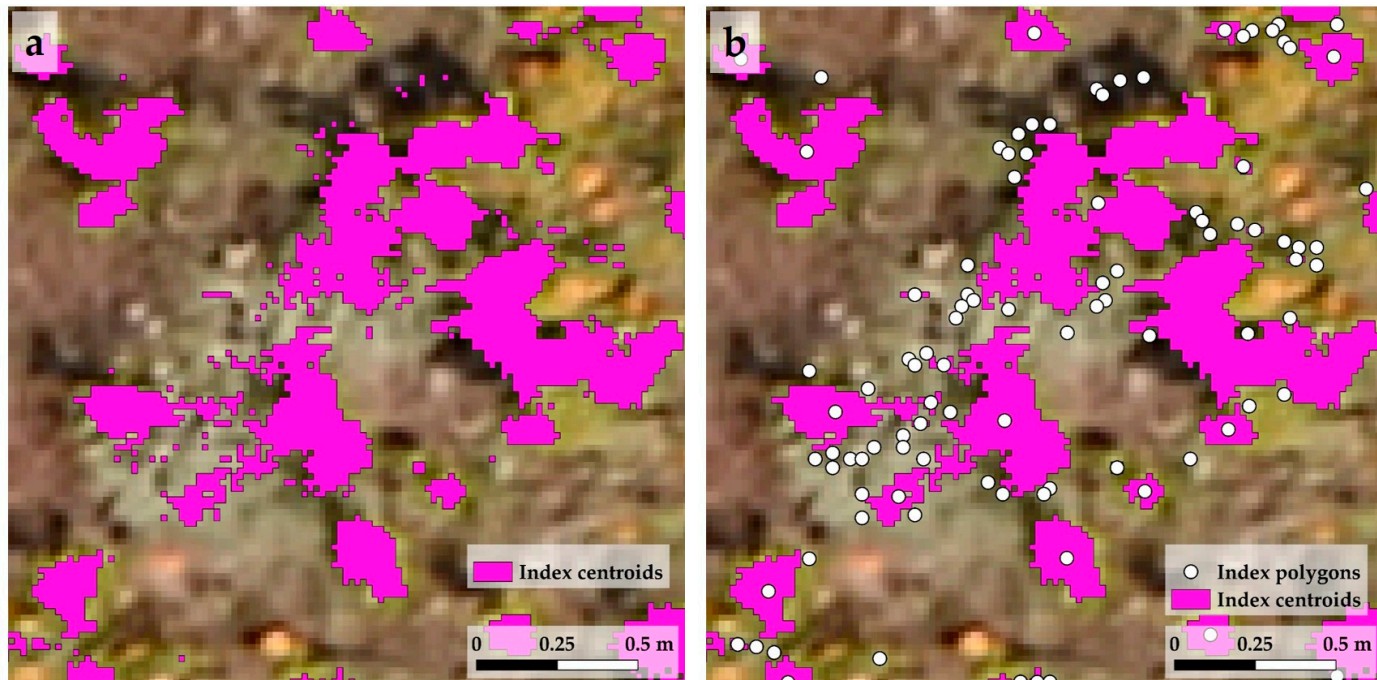

**Figure 8.** The degree of fragmentation of the index-based delineation in the milkweed shoots of index polygons (**a**) and their index centroids (**b**). The images show a part of the first study area of the first stand, the R-G index was used for the delineation in the case of milkweed shoots.

During the accuracy assessment, when comparing the number of pixels of the manually delineated vegetation with the binary maps, the following results were obtained. In the case of milkweed shoots, the best classification could be achieved with the TGI. On average, the Kappa coefficient was 0.71, while the producer's and user's accuracies were 76.38% and 75.42%. This was followed by SSI, whose Kappa coefficient was also 0.71, and the producer's and user's accuracies were 67.02% and 75.12%. Among the two simple indices, G-B gave the best approximation with a Kappa coefficient of 0.69, while the producer's accuracy was 65.62% and the user's accuracy was 73.17% (Table 7). In the case of milkweed, the overall accuracy and errors resulting from the false negative (commission error) and false positive (omission error) classifications are presented in Table S4.

**Table 7.** The confusion matrix of the classification of milkweed shoots. The producer's and user's accuracy values, and Kappa coefficient were used to evaluate the classification results of the pixels of the remotely sensed images. SD, standard deviation.

| | R-G | | | G-B | | | TGI | | | SSI | | |
|---|---|---|---|---|---|---|---|---|---|---|---|---|
| | Producer's acc. % | User's acc. % | Kappa co. | Producer's acc. % | User's acc. % | Kappa co. | Producer's acc. % | User's acc. % | Kappa co. | Producer's acc. % | User's acc. % | Kappa co. |
| STAND I. | 51.51 | 63.53 | *0.57* | 57.45 | 71.66 | 0.63 | 59.62 | 74.31 | 0.66 | 59.44 | 74.27 | 0.66 |
| STAND II. | 65.67 | 66.67 | 0.66 | 73.79 | 74.67 | 0.74 | 75.14 | 76.52 | *0.76* | 74.96 | 75.97 | 0.75 |
| Average | 58.59 | 65.10 | 0.62 | 65.62 | 73.17 | 0.69 | 67.38 | 75.42 | 0.71 | 67.20 | 75.12 | 0.71 |
| SD | 10.01 | 2.22 | 0.06 | 11.56 | 2.13 | 0.08 | 10.98 | 1.56 | 0.07 | 10.97 | 1.20 | 0.06 |

The pixel-based classification of the inflorescences of the blanket flower showed generally worse results here than for the milkweed. The proportion of pixels corresponding to the target vegetation obtained with the structured IF index was the highest: the Kappa coefficient was 0.47, the producer's accuracy was 43.74%, and the users' accuracy was 51.4%. Among the two simple indices, R-G could be used to sort out the pixels belonging to the blanket flower with a Kappa coefficient of 0.4, a producer's accuracy of 36.8% and a user's accuracy of 43.7% (Table 8). Table S5 shows the overall accuracy, false negatives (commission error) and false positives (omission error) obtained during the blanket flower classification.

**Table 8.** The confusion matrix of the classification of blanket flower inflorescence. The producer's and user's accuracy values and the Kappa coefficient were used to evaluate the classification results of the pixels of the remotely sensed images. SD, standard deviation.

| | R-G | | | R-B | | | IF | | |
|---|---|---|---|---|---|---|---|---|---|
| | Producer's acc. % | User's acc. % | Kappa co. | Producer's acc. % | User's acc. % | Kappa co. | Producer's acc. % | User's acc. % | Kappa co. |
| STAND I. | 34.54 | 47.96 | 0.40 | 45.90 | 45.90 | 0.39 | 41.02 | 56.02 | 0.47 |
| STAND II. | 39.06 | 39.44 | 0.39 | 39.35 | 39.35 | 0.39 | 46.46 | 46.77 | 0.47 |
| Average | 36.80 | 43.70 | 0.40 | 42.62 | 42.62 | 0.39 | 43.74 | 51.40 | 0.47 |
| SD | 3.20 | 6.02 | 0.01 | 4.63 | 4.63 | 0.00 | 3.85 | 6.54 | 0.00 |

Overall, the number and ratio of pixels identified as target vegetation showed similar results (Tables S4 and S5, Tables 7 and 8) as the polygon-based comparison. The more structured indices found the target vegetation better than the simpler indices. It is worth mentioning that, in the case of milkweed shoots, larger standard deviations were observed for the producer's accuracy and user's accuracy (Table 7). In contrast to this, there were no significant differences in these parameters between the two stands for blanket flower inflorescences (Table 8).

## 4. Discussion

In the present study, six vegetation indices were examined, calculated from RGB images were examined to provide the best opportunity to classify two selected invasive species.

The values of the index maps calculated from RGB images are typical for the given index. They are in about the same range of values for the study areas. However, the two stands show typical differences within the index value range (Table S1).

The RGB indices have a characteristic threshold value, below or above which the two species can be clearly distinguished from the surrounding vegetation (Table 1). Thus, here we demonstrated that the spectral discrimination of the two invasive plants is also possible with RGB indices. However, these threshold values—within a given index—show differences between the 20 study areas; therefore, the threshold values need to be used as an interval (Table 1). It seems that the threshold values of some indices (R-G and IF) are also affected by the two stands. This dependence can be caused by the time of day, the presence of other similar species, or temporary camera failure, etc. Interestingly, among the simpler indices, the R-G index seemed suitable for both invasive species at the same time for their selection; however, overall, the simpler RGB indices are characterized by high uncertainty and margin of error. In QGIS, the structured indices (TGI, SSI, and IF) are more suitable for selecting the cover of shoots and inflorescences of the two target species. This may be due to the fact that the values of the more structured indices have a much larger interval, and are therefore able to give a finer delineation (wider threshold setting) (Table S1).

In the case of determining the cover of the common milkweed, two indices, TGI and SSI, gave a better approximation, which was supported by the polygon and pixel based validations as well. In the present study, the producer's accuracy of these RGB indices were 76.38% (TGI) and 67.02% (SSI) (Table 7). The accuracy of TGI turns out to be slightly better than the best producer's accuracy obtained during the work of Papp et al. (2021) [63], which was 73.6%. In their work, they used a hyperspectral camera, and SVM and ANN models to identify milkweed plants. The study of Ozcan et al. [64] can also be mentioned, in which a vehicle-mounted camera (RGB) was used to identify milkweed plants autonomously. Among the models they used, ResNet-based Faster R-CNN for milkweed identification achieved a mean average accuracy of 0.98 (training dataset) and 0.44 (test dataset). In contrast, the overall accuracy corresponding to these values in the present study was 99.46% (Table S4). However, the results of the overall accuracy should be treated with caution, as it can be misleading due to the many correctly classified background pixels (Table S4).

Similarly, in the work of Papp et al. (2021) [63], it was also observed that the detection of the milkweed shoots were minimally disturbed by the shrubs of invasive trees that rarely occur in the area (*Ailathus altissima* (Mill.) Swingle and *Robinia pseudoacacia* L.), as they have similar spectral properties to the milkweed shoots. However, these can be excluded, because the height of the shrubs is different from the native vegetation and target plants. They can be excluded by using a digital surface model (DSM) [65], or based on polygons of the nontarget vegetation. Furthermore, they show a lower sensitivity to some species of the open sand grasslands, and the resulting errors can be corrected by optimally setting the threshold value. Furthermore, the two classification methods used by Papp et al. (2021) [63] gave false positive results for bare sandy soils as well, but such false sensitivity was not observed in the present study at all.

Interestingly, in the present study, SSI could only be applied to milkweed shoots and did not give an evaluable signal for blanket flower. This is similar to what Wijesingha et al. (2020) [16] showed in their study of invasive *Lupinus polyphyllus* Lindl. by means of plants in flowering and vegetative state. They solved the problem by extending their methods with two more indices and object based image analysis (OBIA).

It has been shown that hyperspectral RS data can be used to identify milkweeds on UAV aerial photos, but this requires more expensive cameras and more complex processing

methods (e.g., deep learning programs) [63]. Furthermore, Kunah and Papka (2016) [66] investigated the vegetation preferences of milkweeds in agricultural areas using the vegetation indices of multispectral satellite images (including NDVI). This can also be used during surveying with a drone, as the NDVI or SAVI values of the vegetation of natural habitats can be examined with a higher resolution. In this way, it is possible to examine the spatial and temporal changes in the cover of the invasive species and native vegetation or their response to nature conservation treatment. It can be especially useful if the goal is to detect changes in vegetation after the disappearance of invasive species as a result of a nature conservation treatment, which could not be detected by traditional vegetation monitoring [67,68].

In the case of blanket flower, in the RGB images of the areas, it is difficult to separate the vegetative parts of the species (leaf, stem) from the surrounding vegetation, but the inflorescences seem to be more easily recognizable and have a distinctive appearance. The results of the polygon-based validation of the blanket flower fell between the producer's and user's accuracy values of the pixel-based (confusion matrix) validation (Tables 4 and 8). Similar to milkweed, the more structured IF index seemed to be the most sensitive to the area of the blanket flower inflorescences among the RGB indices used in the present study. The best result was given by the user's accuracy of the IF index, but this too barely exceeded 51% (Table 8). In de Sá et al. (2018) [18], the Kappa value was 0.85 for the detection of the yellow inflorescence of *Acacia longifolia* (Andrews) Willd. Similarly to this, Paz–Kagan et al. (2019) [30] achieved a high Kappa coefficient value (0.89) by SVM classification of multispectral images of other yellow-flowered acacia species. In contrast, in the case of blanket flower this value was only 0.47 for the IF index (Table 8). This may be due to the fact that the mass or visibility of the bloom can affect the success of the validation. Due to the shading of the canopy [32] or after the flowering peak, a lower accuracy was achieved when examining the inflorescence of *A. longifolia* [18]. The nonhomogeneous color of the inflorescences can make it more difficult to identify the inflorescences of blanket flower, causing a worse validation result. In general, the centre of the inflorescence of the blanket flower is purple, while the outer edge is yellowish. One solution to this could be the method used by Hill et al. (2017) [13], who created a specific size buffer zone around each pixel identified as a flower. Simpler indices can be used with lower efficiency. This is especially true for the R-B index, which seemed suitable for the simultaneous selection of both invasive species. However, it can be said that in the case of these indices, we did not find flowers or inflorescences of other species, which would cause confusion (perhaps inflorescences of *Targopogon* sp.). Due to the striking color of flowers and inflorescences, even simple RGB images and indices calculated from them can be suitable for their identification, which is regularly used in studies of this kind. Similarly, de Sá et al. (2018) [18] found that the yellowish flower color provided greater specificity in the present study as well. From the point of view of identifying yellow flowers, the blue band is the most important, since the detection of yellow flowers with a low blue value is possible with a combination of red and green [18]. The detection of inflorescences can be useful approaches to study the flowering dynamics of species [30] as a means of calculating reproductive success or propagule pressure. This is because the magnitude of flowering will provide an important indicator of seed productions. Moreover, the numbers, density, and spatial distributions of the target plants can be concluded from this and this can optimize conservation management in a site-specific way, as explained by Gröschler and Oppelt [69]. Blooming can also be used for the detection and mapping of many invasive plant species, as their flowers often have very striking, bright colors, and RGB aerial imagery can be sufficient for this. Nonetheless, floral resource mapping is currently still rarely used in UAV base studies. Carl et al. (2017) [70] successfully used RGB images taken from a drone to not only identify the flowers of black locust (*Robinia pseudoacacia* L.), but also to estimate their number. In the present study, the determination of the area and number of inflorescences was low (Figure 7, Tables 4 and 8). In spite of that the R-B, R-G and IF examined here can also be used for other invasive species with yellow or purple-yellowish flower color, e.g.,:

*Helianthus* agg. or *Solidago* sp. The RGB-based acquisitions of goldenrod species from an airplane were carried out by Bakó (2013, 2015) [7,71]. De Sá et al. (2018) [18] published an interesting example of drone monitoring of flower resources. In their work, *A. longifolia* was used to map the flowering of invasive plant species and to assess the effectiveness of biocontrol treatment. In the case of flowers or inflorescences monitoring, another interesting opportunity may be offered by sensors which is measuring in the ultraviolet (UV) band (280–400 nm) and to choose indices using this band for processing, since many flowers or inflorescences show a very significant reflectance in this band because of the interaction with pollinators [72].

The methods presented in this study gave a good approximation of the area of shoots and inflorescences, but less so for their quantity. It was not possible to determine the exact number of individuals, since the centroids fit into the area of the polygons, but the calculations can even include several shoots or inflorescences in one polygon, if they are located very close to each other. Similar problems occurred with the models of Ozcan et al. (2020) [64], who also observed that they combined the milkweed shoots that were located close to each other. This can cause problems, especially in those parts where the shoots and inflorescences are densely located; therefore the determination of the exact quantity of shoots is limited in such cases. In order to overcome this problem, even for both species, more advanced classification procedures can be applied. The other problem is that index-based delineation resulted in a very fragmented delimitation (Figure 8). That is, one shoot or inflorescence was delimited by several polygons, which thus slightly underestimates the cover (polygons, and pixels) (Figure 6, Tables 3 and 4), but significantly overestimates the number of individuals (centroids) (Figure 7, Tables 5 and 6). One suitable method might be the use of a convolutional neural network (CNN), which could automatically recognize not only the species, but also could separate the individual shoots and inflorescences from each other, thereby giving more accurate plant counts for larger areas. In this study, the use of indices (especially the more structured indices) based on RGB images can be suitable for some basic investigations such as coverage estimations for larger stands of the two invasive species, which, when extrapolated over a longer period of time, can even be used to estimate population or flowering dynamics.

## 5. Conclusions

The study examined the role of six vegetation indices in the identification of two invasive plant species, calculated from RGB drone images in QGIS software. The RGB-based method presented here can provide a good approximation for common milkweed cover and match their location with great accuracy. However, the indices used to separate for the inflorescences of the blanket flower showed a significantly lower percentage of accuracy. Overall, the study shows that RGB indices can be used to identify invasive plants and determine their cover and level of invasion. Thus, it helps the nature conservation treatments, because the method creates an opportunity for practitioners to easily monitor and choose a treatment method based on the density of the target vegetation (e.g., mechanical, chemical treatment). However, due to differences in threshold values, the uncertainty and error of the indices may be high, and the method is less suitable for estimating accurate population quantities. Further development of the results and further refinement of the camera and processing software promote the efficiency and effectiveness of this type of research.

**Supplementary Materials:** The following supporting information can be downloaded at: https://www.mdpi.com/article/10.3390/drones7030207/s1, Figure S1: The correlation between the Manual Polygons (manually delineated polygons) and Index Polygons in the case of the two target species.; Table S1: RGB based indices intervals used to generate binary maps and to which species a given index can be applied.; Table S2: The area of common milkweed shoots area per study areas.; Table S3: The inflorescence area of blanket flower per study areas.; Table S4: The confusion matrix of the classification of milkweed shoots.; Table S5: The confusion matrix of the classification of blanket flower inflorescence.

**Author Contributions:** Conceptualization, L.B., J.S. and P.S.; methodology, L.B. and J.S.; software, J.S.; validation, L.B., J.S. and Z.T.; formal analysis, L.B., J.S. and Z.T.; investigation, L.B. and J.S.; resources, J.S. and P.S.; data curation, L.B., J.S. and Z.T.; writing—original draft preparation, L.B. and J.S.; writing—review and editing, L.B., J.S., C.B., Z.T., B.v.L. and P.S.; visualization, L.B.; supervision, J.S., B.v.L. and P.S.; project administration, J.S. and P.S.; funding acquisition, L.B., J.S., C.B., Z.T., B.v.L. and P.S. All authors have read and agreed to the published version of the manuscript.

**Funding:** This research was funded by National Research, Development and Innovation Office of Hungary, grant number NKFI-6 K124648—"Time series analysis of land-cover dynamics using medium and high-resolution satellite images" project, and by grant TUDFO/47138-1/2019-ITM of the Ministry for Innovation and Technology, Hungary and the WATERatRISK project (HUSRB/1602/11/0057) and the Ministry of Human Capacities, grant number NTP-NFTÖ-20-B-0008.

**Data Availability Statement:** Not applicable.

**Acknowledgments:** The authors also thank the anonymous reviewers for their helpful critical comments and advice to improve the manuscript.

**Conflicts of Interest:** The authors declare no conflict of interest. The funders had no role in the design of the study; in the collection, analyses, or interpretation of data; in the writing of the manuscript, or in the decision to publish the results.

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
