# Peer review of "Drone-Based Identification and Monitoring of Two Invasive Alien Plant Species in Open Sand Grasslands by Six RGB Vegetation Indices"

_drones, doi:10.3390/drones7030207_

Round 1

Reviewer 1 Report

Dear collguess, some minor notices:  (please correct them)!

Line 153 ha -> hectare

Fig 1. right map, north from Izsák undefinied "green lines"=water canals? (not in legend!)

Line 186 Stand I. II. --- sure? explain/correct!

- hy the difference of ~1min/116 pictures???, integration time (pict/sec or speed of UAV ??)

Line 189 "de week" ??? (_when_ was the flight (AM/PM, time ?)

Line 689 "...Efition..." sure?

Author Response

Responses to the Reviewer1

Dear collguess, some minor notices:  (please correct them)!

Response: Dear Reviewer1, thank you for drawing our attention to the errors, thereby improving the quality of the manuscript. We have repaired them.

Line 153 ha -> hectare

Response: Thank you, we use it as you mentioned. (L 188)

Fig 1. right map, north from Izsák undefinied "green lines"=water canals? (not in legend!)

Response: Thank you, we added “Canals” into the legend. (L 204-205)

Line 186 Stand I. II. --- sure? explain/correct!

Response: Thank you for the clarification! We corrected this sentence: “The applied flight parameters for the two stands were: flight altitude of 53.1 m/AED, flight time of 14 minutes and 51 seconds for stand I. and 16 minutes and 1 seconds for stand II.” (L 221-223)

- hy the difference of ~1min/116 pictures???, integration time (pict/sec or speed of UAV ??)

Response: The average flight speed was 13 m/s. (L 219)

Line 189 "de week" ??? (_when_ was the flight (AM/PM, time ?)

Response: We corrected it. “the week” (L 226)

Line 689 "...Efition..." sure?

Response: Thank you, this was a typo. We corrected it (“Edition”) and checked the whole Reference list too. (L 880)

Reviewer 2 Report

Reviewers Comments

The manuscript id: Drones-2234638, titled, “Drone Based Identification and Monitoring of Two Invasive Alien Plant Species in Open Sand Grasslands by Six RGB Vegetation Indices” is well written. The presented work is highly significant in the contemporary context for eliminating plant species that degrade soil quality, but there are some issues that need to be fixed. The author must revise the manuscript and submit a response to the comments for further assessment. Following are specific comments:

1.      In the entire manuscript as well the title, the author must avoid unnecessary capitalization. The entire paper needs to be reviewed and updated.

2.      1-2 sentences of the major findings with numbers should be added in the abstract (the contents at line no 23-26 can be replaced).

3.      Author must avoid the use of first person, intrusion of the first person leads to distraction from the contents and can distort the message.

4.      The current discussion is only a literature review, with no comparison or justification for the obtained results. The discussion should be rewritten with an emphasis on the outcomes provided by the used approaches, which should be compared to the quality of newly released methods with similar functionality. In addition, it is expected that the authors will analyse the results and provide detailed justifications with the aid of relevant literature if odd outcomes are seen.

5.      The authors must rewrite the conclusions (max 150 words), and should present only the major takeaways backed with the numbers. The authors must avoid the sentences presented in line no 554 onwards in the conclusions.

I wish authors a great success.

Author Response

Responses to the Reviewer2

The manuscript id: Drones-2234638, titled, “Drone Based Identification and Monitoring of Two Invasive Alien Plant Species in Open Sand Grasslands by Six RGB Vegetation Indices” is well written. The presented work is highly significant in the contemporary context for eliminating plant species that degrade soil quality, but there are some issues that need to be fixed. The author must revise the manuscript and submit a response to the comments for further assessment. Following are specific comments:

Dear Reviewer2, thank you for your very detailed and helpful comments. You can find our answers to these below.

  1. In the entire manuscript as well the title, the author must avoid unnecessary capitalization. The entire paper needs to be reviewed and updated.

Response: Thank you, we corrected them through the MS (in the Tables and Figures as well).

  1. 1-2 sentences of the major findings with numbers should be added in the abstract (the contents at line no 23-26 can be replaced).

Response: Thank you for the suggestion, we replaced these sentences with some main results. (L 24-30)

.3.      Author must avoid the use of first person, intrusion of the first person leads to distraction from the contents and can distort the message.

Response: Thank you for the comment. We have made the requested changes and we removed the first person from the manuscript.

  1. The current discussion is only a literature review, with no comparison or justification for the obtained results. The discussion should be rewritten with an emphasis on the outcomes provided by the used approaches, which should be compared to the quality of newly released methods with similar functionality. In addition, it is expected that the authors will analyse the results and provide detailed justifications with the aid of relevant literature if odd outcomes are seen.

Response: Some unnecessary parts have been removed from this paragraph. We compared our results with literature data.

  1. The authors must rewrite the conclusions (max 150 words), and should present only the major takeaways backed with the numbers. The authors must avoid the sentences presented in line no 554 onwards in the conclusions.

Response: We took your advice and completely rewrote the Conclusion section. We tried to write this briefly and comprehensively. (L 700-728)

I wish authors a great success.

Thank you, we hope that our manuscript has become more appropriate after the changes made as you and the other reviewers suggested.

Reviewer 3 Report

At present, UAVs with different types of optical sensors have been applied for the detection of different invasive plants in ecosystems, with excellent results. Therefore, the proposed topic is novel. However, there are different recommendations that should be considered:

Summary: It does not clearly show the results of the investigation. Quantitative aspects (values) of the parameters used to assess the quality of the fit with respect to the area of coverage of invasive species should be included. The last sentence “Moreover, our results pointed out that the applied indices can be suitable for use in deep learning programs, which can enable more efficient identification of invasive species”, is a personal opinion, and not a result of the investigation, so it should not be included.

Introduction: The state of the art on the application of UAV and novel procedures for the detection of invasive plants is not addressed. A review of the last three years would be advisable. In addition, a paragraph should be included to summarize the main contributions of your study. Finally, it is recommended to present the last paragraph with a clear structure on the organization of your work.

Materials and Methods: Although the methodology followed in the research is explained, the novelty of the research is not clear. There are different current and efficient procedures that could be considered to improve the results of statistical parameters that are not high, for example classification methods, neural networks, etc.

Manual delineation of polygons using geographic information system software could be automated and would improve accuracy and reduce processing time.

The criteria used to identify the threshold values of the indices for the identification of outbreaks are not clear.

It is advisable to calculate the mean square error, coefficient of variation and other parameters that show the magnitude of the differences between the real and estimated polygons for the invasive plant species, and the same for the centroids.

It is advisable to define a level of significance for the entire investigation, this because different values appear in the document.

Results: The overlap values obtained are low, considering that none exceeds 70% for milkweed and 50% for the inflorescence of the blanket flower, so it would be recommended to reconsider the procedure followed.

Tables 5 and 6 show the percentages of each centroid, which are highly variable from one site to another, so it would be advisable to analyze the procedure followed or consider some modification.

Discussion: There is a lot of information that should be reconsidered, as it does not contribute significantly to the analysis of the results, for example, lines 437 to 450.

It is recommended to review and deepen the explanation of the results obtained and support this discussion with bibliography related to the research. This section mentions possible solutions to the problems encountered, some of which could be implemented to improve the quality of the statistical parameters.

Conclusions: It is recommended to review the wording referring to the results obtained, when recommending the methods based on RGB images, when the results do not support this assertion. It should be considered a main summary of the work.

References: They must be written according to the editorial standards of the journal.

Author Response

Responses to the Reviewer3

At present, UAVs with different types of optical sensors have been applied for the detection of different invasive plants in ecosystems, with excellent results. Therefore, the proposed topic is novel. However, there are different recommendations that should be considered:

Dear Reviewer3, Thank you for your constructive criticism and improvement comments. Our answers to these are as follows:

Summary: It does not clearly show the results of the investigation. Quantitative aspects (values) of the parameters used to assess the quality of the fit with respect to the area of coverage of invasive species should be included. The last sentence “Moreover, our results pointed out that the applied indices can be suitable for use in deep learning programs, which can enable more efficient identification of invasive species”, is a personal opinion, and not a result of the investigation, so it should not be included.

Response: Thank you for the suggestion, we have removed the mentioned problematic claim (L35-37) (and some others) from the summary. And we supplemented the summary with some concrete results. (L 24-30)

Introduction: The state of the art on the application of UAV and novel procedures for the detection of invasive plants is not addressed. A review of the last three years would be advisable.

Response: A review of the literature on remote sensing of invasive species over the last 10-15 years has been carried out: platforms, sensors, processing methods. (L 99-106)

In addition, a paragraph should be included to summarize the main contributions of your study. Finally, it is recommended to present the last paragraph with a clear structure on the organization of your work.

Response: Thank you for the suggestions, we supplemented the manuscript with the main contribution of the study (L 110-120), and clarified the last paragraph in light of the goals of our study (L 122-142). We hope that the aims of our manuscript will become better understood for the readers.

Materials and Methods: Although the methodology followed in the research is explained, the novelty of the research is not clear. There are different current and efficient procedures that could be considered to improve the results of statistical parameters that are not high, for example classification methods, neural networks, etc.

Response: In this study, we present a novel method that is transparent, robust and does not require additional input features. Our method also allows for explanation of the results, which is sometimes difficult for classification results of neural networks, where the values of weights of the model cannot directly be used of interpretation. Finally, the method does not require extra functionality from the software packages that were used.

Manual delineation of polygons using geographic information system software could be automated and would improve accuracy and reduce processing time.

Response: The automatic delineation methods (e.g. the region growing method) did not work properly due to the heterogeneity of the objects, the expertise and manual work of the ecologist was needed to delimit the plants on the reference map and to produce the most accurate reference map possible.

The criteria used to identify the threshold values of the indices for the identification of outbreaks are not clear.

Response: We have supplemented the manuscript with the criteria used to determine the threshold values. The Threshold values were determined as follows: Based on the comparison of manually delineated polygons with index polygons created using index-based selection in QGIS. Six RGB indices were used to examine their applicability in separating the two invasive species, and the value ranges of each index were analyzed to determine the characteristic intervals and sensitivity to the target species. For each study area, threshold values were defined to create binary maps from the index maps that most closely matched the areas of the manual polygons. Overall, the threshold values were selected based on the value ranges of each index and the desired accuracy of the index polygons compared to the manual polygons. (L 313-321)

It is advisable to calculate the mean square error, coefficient of variation and other parameters that show the magnitude of the differences between the real and estimated polygons for the invasive plant species, and the same for the centroids.

Response: We checked the results with a confusion matrix. This method is commonly used to verify the processing of remote sensing data. With this, it was possible to show a slightly better accuracy in the case of the method used. (L 350-366)

It is advisable to define a level of significance for the entire investigation, this because different values appear in the document.

Response: Thank you for drawing our attention to this. In the manuscript, significance is considered from a value of p<= 0.05. (L 373-374)

Results: The overlap values obtained are low, considering that none exceeds 70% for milkweed and 50% for the inflorescence of the blanket flower, so it would be recommended to reconsider the procedure followed.

Response: Our validation method was quite strict. Therefore, as you recommended, we supplemented the checking with another frequently used one (this was the confusion matrix). With this, we were able to show a higher accuracy in the case of our method (the best producer's accuracy of 76.38%, user's accuracy of 75.42% for milkweed shoots). Which is more competitive when compared with literature data. In the case of the blanket flower, this did not prove to be sufficient either (the most appropriate index for IF was 43.74% (producer's accuracy), 51.4% (user's accuracy)). (L 507-538)

Tables 5 and 6 show the percentages of each centroid, which are highly variable from one site to another, so it would be advisable to analyze the procedure followed or consider some modification.

Response: The background of this is that the variation in the number of plants in the stands is natural. Simply put, there are areas where in reality there are more plants and there are areas where there are less. Therefore, the number of manual centroids also differs per stand and the study area as well, as does the number of centroids created for the index polygons. However, the reason for the inaccuracy (between the number of manual centroids and the number of index centroids) is that demarcation with indexes causes fragmentation (Figure 8).

Discussion: There is a lot of information that should be reconsidered, as it does not contribute significantly to the analysis of the results, for example, lines 437 to 450.

Response: Thanks for the comment, the suggested and some other unnecessary parts were removed from this paragraph. (L 543-559)

It is recommended to review and deepen the explanation of the results obtained and support this discussion with bibliography related to the research. This section mentions possible solutions to the problems encountered, some of which could be implemented to improve the quality of the statistical parameters.

Response: We have also supplemented the Discussion section with a comparison of existing literature data and our results. (e.g. L 579-591 and L 627-640)

Conclusions: It is recommended to review the wording referring to the results obtained, when recommending the methods based on RGB images, when the results do not support this assertion. It should be considered a main summary of the work.

Response: We rewrote the Conclusion section based on the results. We tried to write this briefly and comprehensively. We formulated some recommendations based on the present research. (L 700-728)

References: They must be written according to the editorial standards of the journal.

Response: Thank you for the suggestion. We checked the references and corrected them as expected in the Journal.

Round 2

Reviewer 3 Report

The remarks made to the original manuscript were satisfactorily addressed.